# MHC1-TIP enables single-tube multimodal immunopeptidome profiling and uncovers intratumoral heterogeneity in antigen presentation

Mayukha Bathini[1], Diana Bocaniciu[1], Fraser D. Johnson [1], Rick C. P. de Jong [1], Fengchao Yu [2], Vincenzo Davide Aloi[1], Marije C. Kuiken[1,3], Janniek R. Mors [1,3], Lisanne Giebel[1,3], Julien Champagne [1,3], Onno Bleijerveld [1], Reuven Agami [1,3], Krijn K. Dijkstra [1,3], Daniela S. Thommen [1,3], Alexey I. Nesvizhskii[2,4] & Rik G. H. Lindeboom [1] ✉

Profiling antigens presented on MHC class I molecules on the cell surface is essential to identify candidate antigens for targeted and personalized immunotherapies. However, mass spectrometry-based immunopeptidomics has traditionally been limited by high input requirements, extensive sample manipulation, and expensive reagents. To overcome these challenges, we developed MHC1-TIP: a scalable, single-tube and cost-effective workflow to enable robust MHC-I ligandome recovery from cell lines, patient-derived organoids, and sub-milligram amounts of ex-vivo tumour fragments. Moreover, MHC1-TIP also preserves compatibility with additional omics profiling technologies and we demonstrate its capacity for quantitative and multimodal profiling of the proteome and immunopeptidome from the same sample to enable integrated analyses of protein expression and antigen presentation. Application of MHC1-TIP to primary renal cell carcinoma fragments revealed extensive intratumoral heterogeneity in antigen presentation that was poorly correlated with source protein expression. MHC1-TIP represents a broadly applicable and sensitive approach for low-input, multimodal immunopeptidomics with clinical and translational relevance.

Antigen presentation is a critical process that enables the immune system to monitor the proteomic state of cells. In the context of cancer, for instance, T cells can target and eliminate malignant cells by recognizing neoantigens arising from somatic mutations, or self-antigens that reflect aberrant proteomic states characteristic of tumor cells. Accurate measurement of the antigen repertoire is therefore essential for understanding anti-tumor adaptive immunity and prioritizing candidate targets for immunotherapies. However, profiling the immunopeptidome in tissues with limited sample availability remains challenging. Due to technical limitations and low quantities of presented antigens, immunopeptidomics workflows have generally required hundreds of millions of cells, making them poorly scalable and inapplicable to low-input clinical samples[1–3]. Moreover, immunopeptidomics methods tend to use large amounts of pan-MHC class I antibodies for immunoprecipitation of peptide-MHC complexes after cell lysis, rendering them too expensive for routine use.

The lack of scalable, low-input methods for immunopeptidomics is even more of a challenge given that protein expression can be heterogeneous within tumors. If this heterogeneity is reflected within the global antigen repertoire, the associated spatial (and potentially subclonal) variation in antigen presentation may have important implications for immune recognition and the efficacy of antigen-targeted therapies. Understanding how subclonal and topological proteomic differences affect antigen processing and presentation, and how this in turn regionally shapes the antigen repertoire is therefore necessary and requires quantitative, scalable, cost-effective and low-input methods for immunopeptidome profiling. Here, measuring the proteomic, genomic or transcriptomic landscape underlying the antigen repertoire is key for gaining a fundamental understanding of the

[1]The Netherlands Cancer Institute, Amsterdam, Netherlands. [2]Department of Pathology, University of Michigan, Ann Arbor, MI, USA. [3]Oncode Institute, Utrecht, Netherlands. [4]Gilbert S. Omenn Department of Computational Medicine and Bioinformatics, University of Michigan, Ann Arbor, MI, USA. ✉e-mail: r.lindeboom@nki.nl

regulatory mechanisms governing antigen presentation that underpin intratumoral heterogeneity. However, traditional immunopeptidomic workflows do not easily allow for profiling additional modalities as cells are typically lost during sample preparation. This limits the feasibility of performing multi-modal immunopeptidome profiling on small clinical tissue samples. Therefore, methods that enable direct and unbiased measurement of regional variation in antigen presentation in a multi-modal manner need to be developed to study the existence, regulation and implications of intratumoral heterogeneity in antigen presentation.

Apart from the antibody-based workflows, mild acid elution-based immunopeptidomics methods have also been employed for MHC-I ligandome profiling. This method involves treating cells with a mild acid at pH 3 to 3.3, which causes peptides bound to MHC-I molecules on the cell surface to dissociate from the complex and elute. However, previous studies using this method have required large amounts of cells[3,4]. This method also suffers from a large proportion of co-eluting non-antigen peptides that interfere with data acquisition in the mass spectrometer with the classical data-dependent acquisition mode (DDA). In recent years, mass spectrometers have become significantly more sensitive to accurately detect and identify lowly abundant peptides. Hand-in-hand with technological advancement in instrumentation, the proteomics field has begun to more widely adopt data-independent acquisition (DIA) strategies to overcome the limitations of the gold-standard DDA mode. In light of these developments, we sought to leverage the technical advances in mass analyzers, DIA and computational algorithms, and optimize a mild acid elution-based immunopeptidomics experimental workflow for low input samples. Moreover, mild acid elution does not involve sample lysis, making it – in principle – suitable for subsequent multi-omic profiling from the same sample. This resulted in a rapid and scalable workflow, termed MHC1-TIP (**MHC**I **1-T**ube **I**mmuno-**p**eptidomics), which involves minimal sample manipulation, is applicable to cell lines as well as more complex 3D cultures and clinical tissue samples, and can be performed in a single-tube configuration while fitting into a multi-omics pipeline.

## Results

### MHC1-TIP enables scalable and cost-effective immunopeptidomics

To perform immunopeptidomics in a scalable, low-input and cost-effective manner, we first investigated if we could produce a streamlined mild acid elution workflow. Since mild acid elution functions via a single incubation step that specifically elutes MHC-I antigens from the cell surface, we hypothesised that this approach would be uniquely suitable to capture eluted antigens with minimal handling steps. We first evaluated whether mild acid elution indeed represents a suitable method for comprehensive antigen recovery by measuring how much of the MHC-I complexes on the cell surface switch to an open conformation upon mild acid treatment. After a three-minute incubation in the pH 3.3 mild acid elution buffer (adapted from ref.[4]), we stained cells with an anti-pan-HLA-I antibody and measured the signal with flow cytometry in A375 melanoma cells. We found that the mild acid treatment indeed resulted in a strong loss of antibody binding, indicating a loss of intact MHC-I complexes and bound epitopes (Fig. 1A). As a negative control we generated a matched β2-microglobulin (B2M) knock-out line from A375 cells (Supplementary Fig. 1A–C). Without B2M, the MHC class I complex cannot stably fold and load antigens, leading to a total loss in antigen presentation on the cell surface. The signal from the treated cells overlapped with that of the B2M knock-out line, confirming that mild acid elution leads to near-complete loss of surface MHC-I complexes (Fig. 1A and Supplementary Fig. 1D, E) and is appropriate for comprehensive and unbiased peptide recovery. This is consistent with a previously published paper wherein no intact MHC-I complexes were detected by immunofluorescence after mild acid elution[5]. Next, to adapt the mild acid elution method into a single-reaction and a one-tube immuno-peptidomics workflow, we coupled a 10 kDa size exclusion filter to a custom-made 1 mL StageTip (Stop and Go Extraction Tips)[6] containing a C18 resin plug (Fig. 1B). This system combined the observed near-complete

antigen elution upon mild acid treatment with size exclusion filtration to remove cell debris and intact proteins, along with desalting and peptide capture from solution, within a single reaction and pipetting step, enabling an efficient, inexpensive and quick immunopeptidomics workflow that is suitable for low-input samples and does not necessitate any specialised laboratory equipment.

To test whether our workflow, which we termed **MHC**I **1-T**ube **I**mmuno**P**eptidomics (MHC1-TIP), is indeed capable of capturing immunopeptidomes, we investigated the amino acid sequence composition of peptides eluted from 5 million A375 melanoma cells, measured using data-dependent acquisition (DDA). The peptides recovered exhibit features consistent with bona fide antigens (Fig. 1C), including an enrichment of peptides with a length of 9 amino acids (Fig. 1D) and the presence of characteristic anchor residues at position 2 and position 9. To further validate that the peptide characteristics we observed were indeed true antigens dependent on MHC-I antigen presentation, we looked into peptides eluted from 5 million B2M knock-out A375 cells. Reassuringly, the amino acid sequences of peptides eluted from B2M knock-out cells did not display the typical immunopeptidome motif (Fig. 1C), indicating that the MHC1-TIP workflow is indeed capable of capturing a true immunopeptidome.

To assess the purity of the captured immunopeptidomes, we evaluated the percentage of strong and weak MHC-I binders among all detected peptides. We used MHCflurry[7] to predict binding affinity to MHC class I on immunopeptidomes from A375 melanoma cells and JY B-cells generated using MHC1-TIP. The proportion of strong binders (predicted binding affinity lower than, or equal to 100 nM) was slightly below 40% in A375 cells but higher than 65% in JY cells (Supplementary Fig. 2A). This indicates that relative antigen yield is variable between cell types, and is likely in part related to the amount of MHC class I that is expressed. To overcome limitations in purity of mild acid elution-based immunopeptidomes, which have historically suffered from relatively high amounts of co-eluting peptides, we sought to improve MHC1-TIP by leveraging the sensitivity of DIA. For analysis of DIA data, we used the MSFragger-powered FragPipe computational platform to generate a deep spectral library using triplicates of 100 million A375 cells with an immunoprecipitation (IP)-based immunopeptidomics workflow, measured using DDA. This spectral library was then used to search all subsequent immunopeptidomes from the A375 cell line. With this spectral library, we tested the performance of MHC1-TIP in comparison to the IP-based workflow, which is considered the standard in the field, on samples with varying input amounts in DIA mode. Reassuringly, we observed a strong overlap in the antigen repertoire detected by both methods. Using MHC1-TIP, we detected more than 500 antigens at the lowest input of 100,000 cells, while 5 and 10 million cells yielded more than 5000 antigens (Fig. 1E). This is in stark contrast to published work, where typical input amounts required 100 million cells or more to detect similar amounts of peptides[3,4]. We noted that – when combined with our optimized workflows – we were able to identify similar numbers of peptides with MHC1-TIP as with IP-based immunopeptidomics. This indicates that MHC1-TIP in conjunction with DIA achieves a similar depth despite high amounts of contaminant peptides inherent to the mild acid elution approach. We next compared the peptide repertoire obtained with both methods and their abundances. With the 5 million cell samples, we observed that 6443 peptides were detected in both methods, while 803 peptides were found uniquely in the IP method and 171 peptides in MHC1-TIP samples alone. We also observed that intensities of peptides identified with both methods correlate strongly (Fig. 1F), indicating that MHC1-TIP is a quantitative approach for detecting antigen abundances that is on par with IP-based immunopeptidomics, while being more scalable, accessible, quick, and inexpensive.

We then tested the ability of MHC1-TIP to capture quantitative changes in antigen presentation by treating the melanoma A375 cell line with a series of interferon gamma (IFNγ) doses ranging from 5 to 100 ng/mL (Fig. 1G), measuring the immunopeptidomes in DIA and analysing them with the spectral library described above. IFNγ is known to both stimulate

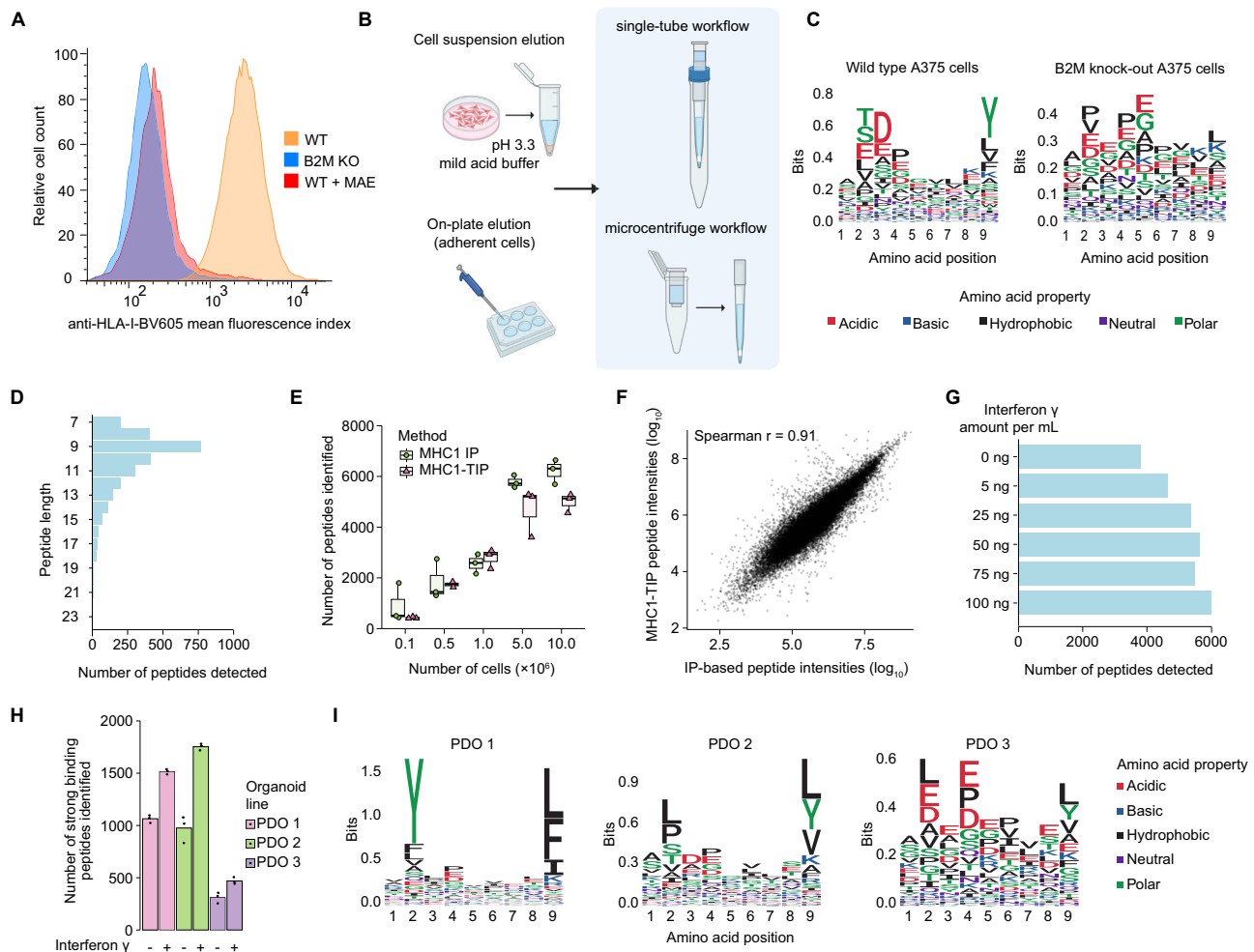

**Fig. 1 | Establishment and validation of MHC1-TIP. A** Representative histogram showing reduction in antibody binding to MHC-I complexes after mild acid treatment (MAE) by flow cytometry with wildtype (WT) and B2M knock-out (KO) A375 cells. **B** Schematic workflow of MHC1-TIP wherein antigens can be eluted off cells in suspension or, for adherent cells, directly from the culture dish. The eluted antigens are then subjected to ultrafiltration using a 10 kDa size exclusion ultrafilter and desalted with a C18 plug in a 1 mL pipette tip, either in a single-tube or microcentrifuge configuration. Created in BioRender. **C, D** Immunopeptidomes were measured using DDA from 5 million A375 cells. **C** Sequence motifs of peptides of length 9, eluted from wild-type or B2M knock-out A375 cells with MHC1-TIP. The motif displayed in the wild-type cells matches that of the MHC-I haplotypes of the A375 cell line (HLA-A*01:01, HLA-A*02:01, HLA-B*44:03, HLA-B*57:01, HLA-C*06:02, HLA-B*16:02). **D** Length distribution of identified peptides. **E–G** Immunopeptidomes were measured in DIA mode and were analyzed using a spectral library generated from IP-based immunopeptidomes of A375 cells.

**E** Number of peptides identified with MHC1-TIP or MHC-I immunoprecipitation (MHC1 IP) with different cell input numbers. The central lines in the shown boxplot are the median, the box shows the interquartile range, and the whiskers are extreme values upon removing outliers. **F** Spearman correlation between immunopeptide intensities quantified with MHC1-TIP and MHC1 IP, using all samples displayed in (**E**). **G** Number of peptides identified with different concentrations of IFNγ in A375 cells with MHC1-TIP, using 1 million A375 cells per dose, (**H**) Number of immunopeptides identified from 3 different colorectal cancer patient-derived organoid (PDO) lines with MHC1-TIP (using DDA), with and without IFNγ treatment. The peptides were filtered for strong binders with predicted binding affinities below 100 nM based on known MHC-I haplotypes of each line. (**I**) Sequence motifs of peptides of length 9, eluted from the 3 PDO lines. The motifs match the known MHC-I haplotypes of the lines (see Methods).

the production of antigens by activating the immunoproteasome, and increase the amount of presented antigens on the cell surface by upregulating MHC genes and other genes involved in antigen presentation. Using 1 million A375 cells per dose, we observed a dosage-dependent increase in detected immunopeptides, plateauing after 25 ng/mL of IFNγ (Fig. 1G), highlighting the ability to detect immunopeptidome dynamics. These experiments also revealed that even at low cell input amounts and without additional stimulation of antigen presentation, MHC1-TIP is capable of detecting several tumor antigens derived from cancer testis antigens that are known to be specific to melanoma cells (Supplementary Fig. 2B).

We next investigated if MHC1-TIP specifically captures antigens eluted from MHC class I, or if co-elution of MHC class II antigens can occur. The A375 melanoma cell line is known to also present MHC class II

antigens, which could be upregulated by the IFNγ treatment. To test the specificity of MHC1-TIP for MHC class I peptides, we measured the same IFNγ-treated A375 immunopeptidomes in DDA mode for an unbiased analysis and used NetMHCIIPan[8] to measure the presence of MHC class II binding peptides (Supplementary Fig. 2C). Reassuringly, we only detected between two and six MHC class II binders in melanoma cells and no correlation between interferon concentration and the amount of MHC class II ligands, suggesting a high specificity of mild acid elution at pH 3.3 in eluting MHC class I peptides, in line with a previously published report[5].

Mild acid elution has – to the best of our knowledge – thus far only been applied to cells cultured in 2D or suspension, presumably because of difficulties isolating large numbers of viable cells from complex 3D culture models and tissues. This has limited the implementation of mild acid elution

in studies that use clinical material, or 3D cell culture models such as patient-derived organoids that more closely resemble tissues. To test if our newly developed MHC1-TIP approach is suitable for such samples, we used three colorectal cancer patient-derived organoid (PDO) lines with known MHC-I haplotypes. These experiments revealed that MHC1-TIP is indeed capable of profiling organoid-derived immunopeptidomes, identifying up to 1000 peptides from individual wells from a 6-well culture plate, which further increased to > 1500 peptides when the organoid cultures were stimulated with IFNy (Fig. 1H and Supplementary Fig. 2D). Reassuringly, the amino acid sequence motif from each organoid line closely matched that of the MHC-I haplotype that was determined for each patient (Fig. 1I), based on the MHC Motif Atlas[9]. Of note, we observed that the use of dispase to dissociate organoid cultures introduced technical artifacts containing the N-terminal cleavage pattern of dispase, likely to be dispase-cleaved peptides; we therefore advise against the use of proteases when harvesting organoids for immunopeptidomics (Supplementary Fig. 2E). These results demonstrate that mild acid elution, and specifically MHC1-TIP, offers a scalable alternative to IP-based immunopeptidome profiling in complex samples.

## MHC1-TIP enables multi-omics profiling from low-input samples

Since MHC1-TIP leaves cells intact while eluting and capturing antigens, we hypothesised that this method would be uniquely suitable for integration with other downstream workflows without compatibility issues, to probe additional layers of information from the same samples in a multi-omics profiling approach. Being able to obtain multimodal omics data is especially important when samples are precious, scarce, limited, or are difficult to culture and expand into sufficient quantities, such as organoids, biopsies, and tissue samples. Moreover, matched datasets from exactly the same samples allow more comprehensive integration of modalities while eliminating batch effects and specimen-to-specimen variability. To verify that cells indeed remain viable for downstream workflows, we investigated the effect of mild acid treatment on the cells' proteomes. Reassuringly, treating A375 cells with mild acid did not lead to major changes to the proteome, where we observed only 8 proteins to significantly change upon mild acid treatment (Supplementary Fig. 3A). We also observed a significant depletion of B2M, as expected after mild acid elution (Supplementary Fig. 3A).

To test if MHC1-TIP is suitable for profiling additional modalities in addition to the immunopeptidome, we performed proteomics on the above-mentioned patient-derived organoid lines with the cells that were retained after antigen elution. We could identify and quantify more than 8000 proteins, underscoring the ability to obtain deep insights into protein expression dynamics in conjunction with the MHC1-TIP workflow. These results also uniquely enabled integrative analyses between protein expression and antigen presentation dynamics. For example, we observed that PDO-3 had significantly lower expression of antigen presentation-related proteins such as the HLA proteins, which coincided with significantly lower amounts of immunopeptides detected in the matched immunopeptidomes. We also observed differences in how the respective lines responded to IFNy. For instance, IRF1 was mainly induced in PDO-2 while the IRF3/6 expression was seen in PDO-1 (Fig. 2A). These results highlight the need for multimodal profiling in combination with immunopeptidomics in patient-derived low-input samples with clinical translatability to gain insights into antigen presentation from a system biology perspective.

Multimodal proteome-immunopeptidome profiling uniquely enables the investigation of how protein abundance relates to the amount of antigen presented from that protein. To explore how changes in protein expression would reflect in the immunopeptidome, we treated the A375 melanoma cell line with tumour growth factor beta (TGFβ) for 18 h and profiled the induced proteomic and immunopeptidomic dynamics. Treated cells showed an upregulation of several proteins involved in TGFβ signalling such as TGFBI, FN1, CCN2 and SMURF2 (Fig. 2B), as well as the epithelial mesenchymal transition and TGF beta signaling gene signatures (Supplementary Fig. 3B). On the immunopeptidomic level, we also observed significant changes in the presentation of several peptides after treatment (Fig. 2C). Interestingly, while 81 proteins were significantly differentially

expressed and antigens with changes in presentation were derived from 85 source proteins, we detected very limited overlap between these two modalities (Fig. 2D). This reveals that most of the significant changes in the immunopeptidome were not driven by changes in the proteome, and indicates that antigen repertoire dynamics are independently regulated, underscoring the need for multi-omic immunopeptidome profiling.

## Immunopeptidome-proteome profiling from low-input primary patient tissues with MHC1-TIP

Finally, we sought to apply the multi-omic MHC1-TIP approach in a clinical setting by profiling antigen presentation in patient-derived tissue fragments. Mild acid elution has thus far not been used on primary human tissues, presumably because it requires live cells in suspension, and is incompatible with freezing. To ensure cell viability and circumvent this limitation, we used a previously published ex vivo tumour fragment culture method[10] to revitalize cells for one day from cryopreserved tissue fragments of a renal cell carcinoma tumour. Moreover, we reasoned that if tissue fragments were small enough (~1 mm$^3$), mild acid should result in the elution of immunopeptides without the need for dissociation of the tissue into a single-cell suspension. Indeed, a five-minute incubation of intact tissue fragments in mild acid resulted in elution of peptides that contained sequence motifs characteristic of the patient's MHC-I supertypes (HLA-A*03; HLA-B*07, HLA-B*08; HLA-C*07) (Fig. 2E) and also displayed an enrichment of peptides of length 9 (Supplementary Fig. 4A). This shows, for the first time, that mild acid elution can be used for reliable and rapid immunopeptidome profiling from primary tissues. Notably, dissociation of the fragments using collagenase, hyaluronidase, and DNase into a single-cell suspension introduced technical artefacts and yielded peptides that appeared to be digestion products, again underscoring the incompatibility of enzymatic dissociation methods with mild acid elution (Supplementary Fig. 4B).

We hypothesized that the compatibility with low input samples and the scalability of MHC1-TIP would make it uniquely suitable for measuring immunopeptidome profiles across different sites within the same tumour. The biobanked tissue fragments that we utilized for this study were resected from different topological regions of the same tumour, and thus suitable to evaluate intratumoural heterogeneity. While heterogeneity in global amounts of MHC class I and the expression of individual tumor antigens within the same tumour has been studied and proposed to contribute to immune evasion[11], the heterogeneous presentation of (tumour) antigens in an unbiased immunopeptidome-wide manner has not yet been explored, likely because of scalability, cost-effectiveness and the amount of sample required for immunopeptidomics workflows in the past. We were able to quantify more than 8000 proteins and a wide range immunopeptides of length 9 from the equivalent of half a tissue fragment (< 1 mm$^3$; estimated to contain < 10,000 to 50,000 cells) (Fig. 2F, G), which alluded to differences in antigen presentation across different regions of the tumour. This highlights the sensitivity and scalability of the MHC1-TIP workflow in capturing antigens from extremely low-input clinical samples and presents a unique dataset with the opportunity to explore immunopeptidomic and proteomic heterogeneity in parallel.

## Data independent acquisition and B2M based normalisation enables quantitative interrogation of tissue samples

In addition to the new experimental workflow that MHC1-TIP represents, we also sought to improve downstream mass spectrometry measurement and quantification. In particular, we investigated if data acquisition, processing and analysis could be optimized to maximize the quantity and quality of antigens identified in low-input and precious clinical samples. While mass spectra for immunopeptidomics have traditionally been collected through data-dependent acquisition (DDA), we explored whether the recent advances in data-independent acquisition (DIA) for proteomics and narrow window selection in conjunction with the Astral mass analyzer could be leveraged to apply DIA on immunopeptidomics. DIA immunopeptidomics data has so far mainly been analysed using DDA-generated

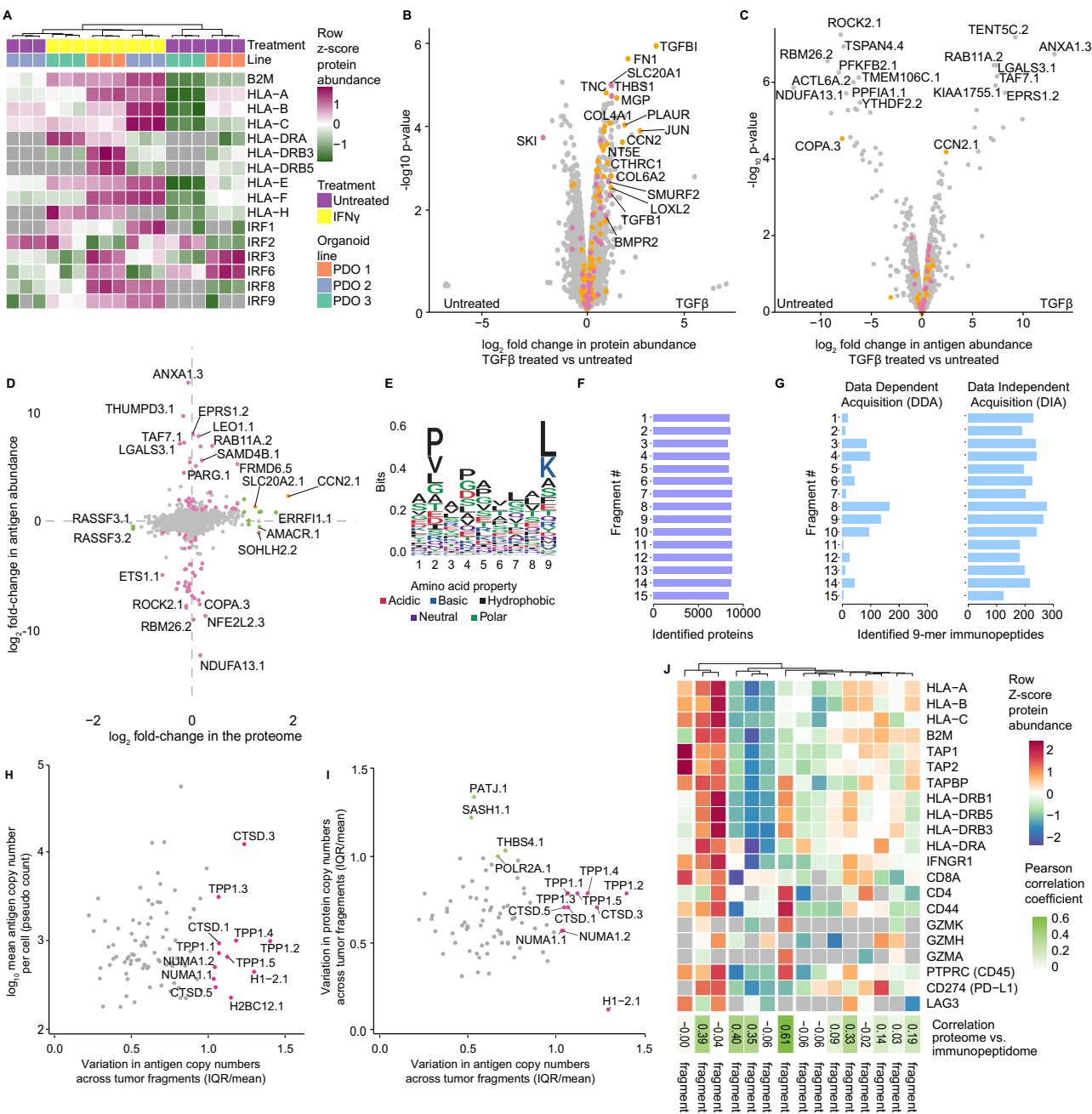

**Fig. 2 | MHC1-TIP enables multi-omic profiling from cell lines, organoids and primary tissues. A** Heatmap displaying expression of proteins involved in antigen presentation in the three colorectal cancer patient-derived organoids (PDO) lines, with and without IFNγ treatment (200 ng/mL for 24 h). The retained cells after immunopeptidome elution were processed for proteomics. **B–D** Proteomic (**B**; $N = 2$ untreated and $N = 3$ treated) and immunopeptidomic (**C**; $N = 3$) changes induced after treatment with TGFβ (5 ng/mL for 18 h). **B, C** Log$_2$ fold-changes are plotted against -log$_{10}$ un-adjusted p-values computed using empirical bayes statistics. Genes belonging to the Hallmark pathways "Epithelial Mesenchymal Transition" and "TGF Beta Signaling" are coloured in orange and pink, respectively. **C, D** The immunopeptidomes were measured in DIA mode and analyzed with a spectral library generated from IP-based immunopeptidomes of A375 cells. Immunopeptides are labelled with the gene name of the parental protein of the antigen, conjugated with a numeric identifier for each peptide identified from that protein. **D** Integrated proteomic and immunopeptidomic changes upon TGFβ treatment. Antigens and proteins that are significantly up- or down-regulated (based on FDR-adjusted p-values < 0.05 and log$_2$ fold-changes > 1; adjusted p-values computed from moderated t-statistics of empirical bayes testing) are coloured in pink and green, respectively. **E** Sequence motif of peptides of length 9 eluted from 15

patient-derived tumour fragments represent the anchor residues characteristic of the known MHC-I supertypes of the patient: HLA-A*03, HLA-B*07, HLA-B*08, HLA-C*07. **F** Number of protein groups identified from each fragment. **G** Number of peptides of length 9 identified from the immunopeptidome of each tumour fragment in data-dependent (DDA) and data-independent (DIA) acquisition modes. **H** Variation in antigen pseudocounts per cell across the 15 tumour fragments. IQR, interquartile range. **I** Variation in antigen pseudocounts per cell in relation to variation in their source protein copy numbers per cell, across the 15 tumour fragments. **H, I** Immunopeptidomes were measured in DIA mode and analyzed with direct DIA. The peptides were then filtered based on their match with the core MHC class I binding motif (using GibbsCluster[18]) and for length 9. Peptides with more than 5 missing values across the 15 fragments were removed. Immunopeptides are labelled with the gene name of the parental protein of the antigen, conjugated with a numeric identifier for each peptide identified from that protein. **J** Heatmap displaying copy numbers per cell of proteins involved in antigen presentation and markers of T cell infiltration in each tumour fragment. The correlation between z-score normalized protein copies and antigen pseudocounts per cell, are shown as the Pearson correlation coefficients within each tumour fragment.

spectral libraries from matched samples. While this approach is useful when studying antigen presentation with immortalized cell lines, it is often not possible to generate an experimental spectral library when sample availability is limited. To overcome this limitation, we explored the direct DIA approach, first proposed in ref. 12, and further advanced in the MSFragger-DIA search engine[13] in the FragPipe computational platform[13–15] to analyse DIA immunopeptidomes without a pre-existing spectral library. To compare direct DIA with data analysis using a pre-existing spectral library generated by MHC1 immunoprecipitation, we re-analysed the DIA MHC1-TIP and IP-based immunopeptidomes from A375 cells. This revealed strikingly similar numbers of strong-binding immunopeptides identified by both methods (Supplementary Fig. 4C). The peptide ion intensities quantified by both data analysis workflows also showed a strong correlation with each other (Supplementary Fig. 4D). When analysing the MHC1-TIP 5 million cell samples, we found that 3721 of the 5802 strong binders identified by direct DIA were also found with the pre-existing spectral library. This demonstrates that MHC1-TIP with DIA attains a depth analogous to that of the conventional MHC-I IP, despite the presence of substantial amounts of co-eluting contaminant peptides, while offering advantages in time and cost efficiency.

We measured the immunopeptides eluted from the 15 fragments in both DDA and DIA mode to test if the direct DIA analysis workflow could improve the detection and quantification of antigens in our tumor fragment samples as well. This revealed a striking improvement in data completeness across the tumor fragments in DIA measurements compared to the conventional DDA method (Fig. 2G, Supplementary Fig. 4E, F), highlighting the capabilities of DIA coupled with the FragPipe platform in generating deep datasets suitable for quantitative studies with limited sample availability.

We next sought to improve the normalization of immunopeptide ion intensities for quantitative comparisons of antigen abundance across the different fragments. We observed a high degree of variability in the sizes of the tumour fragments and, thereby, the number of cancer cells they contain which inevitably biased ion intensities and hindered quantitative comparisons between fragments. To address this, we first normalized antigen intensities by calculating each antigen's fractional contribution to the total signal of all peptides within a sample, preserving stoichiometric relationships while correcting for unequal cell inputs. Next, to estimate antigen copy numbers per cell, we scaled these fractions by the average number of β2-microglobulin (B2M) molecules per cell measured in the matched proteomes of each fragment, based on the assumption that stable surface expression of MHC-I complexes depends on the availability of B2M molecules, which therefore should be directly proportional to the total number of antigen-MHC-I molecules presented on the cell surface. To estimate B2M copy numbers, we used the 'proteomic ruler' approach[16], which infers absolute protein abundances from histone MS signals by leveraging the known DNA content per diploid cell. Applying this method allowed us to derive absolute antigen pseudocounts per cell, removing the bias introduced by variable sample inputs. To validate our approach, we calculated the pseudocounts of antigens quantified in the untreated samples from the TGFβ-perturbation experiment and compared our results to published absolute antigen counts in A375 cells by Stopfer et al. (2022)[17]. Among the 11 antigens they quantified using hipMHC spike-ins, 5 overlapped with our dataset. Our calculated pseudocounts strongly correlated and were in agreement with their measured values (Supplementary Fig. 4G), supporting the quantitative power and accuracy of our absolute antigen normalisation method.

### Tumour fragment immunopeptidomics reveals intratumoral heterogeneity in antigen presentation

Next, we set out to leverage this unique, quantitative integrated immunopeptidome and proteome dataset of patient-derived tumour fragments from different regions of one renal cell carcinoma tumour generated with MHC1-TIP to measure intratumoral heterogeneity in antigen presentation. Clustering peptides by sequence (using the GibbsCluster tool[18]) and selecting biologically relevant MHC-I antigens revealed extensive variation in total antigen abundance. For instance, tumor fragment 5 exhibited the highest levels of presented antigens, whereas tumor fragment 15 displayed minimal presentation (Fig. 2G, Supplementary Fig. 5A, B), suggesting spatial and potentially subclonal differences in antigen processing. To identify antigens that were heterogeneously presented across the tumor, we quantified variability in antigen presentation by calculating the interquartile range (IQR) scaled to the mean abundance of each antigen. This highlighted highly variable antigens, including several peptides derived from TPP1 and CTSD, enzymes involved in extracellular matrix degradation and known to be upregulated in several tumours[19,20] (Fig. 2H). Interestingly, variation in antigen presentation was poorly correlated with the variability in expression of their source proteins (Fig. 2I), and this correlation between protein and antigen abundance varied across fragments (Fig. 2J). This suggests that intratumoral heterogeneity in antigen abundances and repertoire is not explained by underlying variation in protein expression, and that antigen processing and presentation might be independently regulated. This could have important translational implications, for example, when selecting tumor antigens for inclusion in cancer vaccines that are currently selected based on expression levels[21]. In addition, the extensive observed heterogeneity indicates that not all (tumor) antigens are suitable for effective clearance of the entire tumor.

Finally, we used the matched protein expression data that multi-omics MHC1-TIP provides to investigate whether the expression of antigen presentation machinery correlated with immune infiltration markers and with immunogenicity of each fragment. Hierarchical clustering based on MHC (HLA) protein expression, immune infiltration markers, and antigen processing components revealed distinct immunological profiles across fragments (Fig. 2J). Tumor fragment 5 stood out as the most "immune-hot" region, with elevated expression of B2M, MHC-I/II, TAP1/2, and markers of CD8/CD4 T cell infiltration, which in turn correlated with its high antigen abundances. Tumor fragments 1 and 3 also showed relatively higher MHC-I expression and corresponding antigen presentation, although the relatively low expression of B2M in tumor fragment 1 may explain its lower antigen display. Markers of T-cell activation (e.g., CD44, granzymes), exhaustion (LAG3), and immune evasion (PD-L1) were also detected. In contrast, fragment 15 had high immune cell content (CD45) and MHC-II expression, coinciding with high CD4 but not CD8 expression, which could suggest the T cell response is skewed towards the CD4+ subtypes. Notably, LAG3 was highly expressed in both antigen-high (fragment 5 and 7) and antigen-low (fragment 1) regions, suggesting complex regulatory dynamics. Strikingly, fragments categorized as immune-cold (tumor fragments 11 and 12) also clustered together based on their immunopeptidomes, underscoring the sensitivity of our assay in capturing meaningful biological variation from low-input clinical samples.

## Discussion

We present MHC1-TIP, a novel workflow for multimodal immunopeptidomics that offers a reliable, efficient, cost-effective and quick alternative to conventional methods that involve extensive sample processing steps, necessitate large sample inputs and require expensive reagents such as columns and large amounts of antibody and beads. Recent reports have also successfully miniaturized antibody-based immunopeptidomics using microfluidic devices, thereby significantly reducing sample and reagent requirements, however, they require specialized equipment[2,22,23]. MHC1-TIP is a simplified method, based on mild acid elution that is capable of capturing antigens from as low as < 1 mg of tissue containing around $10–50 \times 10^3$ cells at a much reduced cost.

Mild acid elution-based immunopeptidomics has historically suffered from substantial coelution of contaminant biomolecules and peptides that interfere with the identification of true MHC-I binding antigens. We observe variability in the proportion of strong MHC-I binding peptides to coeluting peptides based on the cell type, which is expected to be associated with variability in MHC class I expression. While purity can be improved by starting with high-input cell amounts and implementing additional purification steps (such as sequential centrifugation) as reported by Sturm et al.[4],

we showcase how these limitations can be surpassed in a low-input setting through a combination of methodological and analytical advances. Specifically, we describe how data-independent acquisition (DIA) for immuno-peptidomics can result in more complete data from minimal sample input even without a pre-built experimental spectral library, which improves the sensitivity of MHC-I antigen profiling for clinical applications. We also demonstrate that coupling mild acid elution with DIA, which provides deeper and more reproducible peptide coverage, allows confident detection and discrimination of true MHC-I ligands despite substantial amounts of contaminant peptides through stringent computational filtering based on peptide length constraints, binding predictions, and sequence motif clustering.

Furthermore, to the best of our knowledge, we demonstrate for the first time that mild acid elution-based immunopeptidomics can be applied to complex samples such as organoids and tissues. In our study, we utilized an ex vivo culturing system that enabled the use of cryopreserved and frozen tissues, which also facilitates the diffusion of peripheral blood–derived cells and peptides out of the tissue fragment. However, there could still be some residual contribution to the peptide pool from these components. While we expect MHC1-TIP to also be applicable to fresh tissues, we acknowledge that such contamination could be more substantial. Computational filtering for characteristics of canonical MHC class I peptides, such as peptide length, predicted binding affinities, and sequence clustering can help minimize this issue of contamination in complex tissue types.

Moreover, we also highlight how samples including patient tissues, after stripping away antigens, still remain suitable to measure additional modalities in a multi-omics setting. In our study, we exemplify a number of applications of proteomic profiling in combination with immunopeptidomics, and we expect that this can be extended to parallel genomics and transcriptomics profiling, linking DNA aberrations or transcriptional and translational dynamics with antigen presentation. We observed an unexpected discordance between immunopeptidome dynamics and changes in protein expression of their respective source proteins, both in experimental perturbations and in the context of observed variability across a tumor. The poor correlation between proteomic and immunopeptidomic changes indicates that expression profiling could be a poor predictor for antigen presentation, and that additional layers of regulation dictate the composition of the presented antigen repertoire. Antigens presented on the cell surface come from a pool of proteasomal degradation products that are transported into the ER and loaded onto MHC-I complexes. Increased presentation of an antigen could therefore be regulated by increasing degradation of a protein without altering the steady state abundance of this protein.

MHC1-TIP uniquely enabled the deep interrogation of immuno-peptidomic intratumoral antigen presentation dynamics. This revealed unexpected intratumoral heterogeneity in the composition and presentation of the antigen repertoire across different topological domains of a single tumor. The presence of a subset of antigens, which includes tumor antigens, that are not homogeneously presented across a tumor has important immunological implications for the targetability of the tumor by T cells. This suggests that taking antigen presentation heterogeneity into account is important for the prioritization of candidate tumor antigens selected as therapeutic targets in personalized immunotherapies, such as cancer vaccines. Importantly, observed heterogeneity could not be explained by changes in protein expression, further indicating that expression profiling may not be sufficient for selecting targetable antigens. MHC1-TIP offers a scalable and cost-effective immunopeptidomics workflow, suitable for routine clinical implementation, to profile MHC-I antigens from clinical samples and surpass these limitations in prioritizing candidate antigens for targeted immunotherapies.

## Methods
### Chemicals
All chemicals used were analytical grade, obtained from Sigma-Aldrich, unless mentioned otherwise. LC-grade water, methanol, and acetonitrile were obtained from Biosolve.

### Cell culture
The A375 cell line was cultured and maintained in Dulbecco Modified Eagle Medium (DMEM) (GIBCO) supplemented with 10% FBS (Serana) and 1% penicillin-streptomycin (GIBCO) at 37 °C. For the TGFβ stimulation experiment, the cells were treated overnight with 5 ng/mL of human TGFβ (240-B-002, Bio-Techne) for 18 h. For interferon stimulation, the cells were treated with a range of IFNγ (PHC4031, GIBCO) doses from 5 ng/mL to 100 ng/mL for 24 h. JY cells were cultured in Roswell Park Memorial Institute (RPMI) 1640 (GIBCO), supplemented with 10% FBS and 1% penicillin-streptomycin. Both cell lines were originally obtained from ATCC.

### Flow cytometry
A375 cells were washed with PBS and 150 mM NaCl, followed by a 3 min incubation in pH 3.3 mild acid buffer composed of 131 mM citric acid, 66 mM $Na_2HPO_4$, 150 mM NaCl, and 25 mM iodoacetamide (Cytiva) and adjusted to pH 3.3 with NaOH (adapted from ref.[4]). Cells were then washed in FACS buffer [PBS (GIBCO) supplemented with 1% FBS] and stained with an empirically determined dilution factor of 1:50 of W6/32-BV605 (BioLegend, 311432) anti-pan-HLA antibody, anti-HLA-A2-APC (BD Biosciences, 561341), mouse IgG2b-APC (BioLegend, 402205) isotype control and 1:300 of Invitrogen™ LIVE/DEAD™ Fixable Near-IR Dead Cell Stain (Invitrogen, 15519340) for 20 min at 4 °C, and then washed twice with FACS buffer. After incubation the cells were washed and analysed on the FACSymphony™ A5 (BD Biosciences). Measurements performed from technical triplicate samples were processed in parallel. FCS files were analysed in FlowJo 10.10.0.

### B2M knockout
**Cloning B2M KO vector.** Primers encoding the B2M target sequence (F: CACCGAAGTCAACTTCAATGTCGGA, R: AAACTCCGA-CATTGAAGTTGACTTC) were annealed and ligated (New England Biolabs, M0202L) into a BsmBI-v2 (New England Biolabs R0739S)-digested pLentiCRISPR-mCherry backbone (Addgene plasmid #75161, gift from Beat Bornhauser[24]), according to the Zhang Lab Lentiviral CRISPR Toolbox protocol[25,26]. The resulting ligation product was transformed into Invitrogen™ One Shot™ Stbl3™ Chemically Competent E. coli (10193952) and selected on ampicillin (Fisher Scientific, BP176025) Luria broth (Fisher Scientific, 12795084) agar (Fischer Scientific, BD 214010) plates. Selected colonies were inoculated into LB-ampicillin medium and incubated for 18 h on an orbital shaker at 37 °C. Plasmid DNA was purified using QIAprep Spin Miniprep Kit (Qiagen, 27106) and insertion of the correct target sequence was confirmed following column cleanup (QIAquick PCR and Gel Cleanup Kit) to remove PCR primers, through Sanger sequencing by Mix2Seq Kit NightXpress (Eurofins Genomics Europe) using the LKO.1 5' primer (GACTATCATATGCTTACCGT).

**Generation of A375 B2M KO cell line.** A375 cells were seeded into a 10 cm plate at 50% confluence. The following day, 8 μg pLentiCRISPR-mCherry-sgB2M was transfected by combining with 24 μg linear MW 25000 poly-ethylenimine (Polysciences, 23966) in Opti-MEM reduced serum medium (ThermoFisher, 31985062) and adding dropwise to the adhered cells. Two days later, mCherry-positive cells were single-cell sorted into a 96-well plate. The resulting B2M KO clones were expanded and loss of mCherry expression was confirmed by microscopy, as we sought to confirm the absence of residual pLentiCRISPR-mCherry-sgB2M resulting from transient transfection. Genomic DNA was extracted using MightyPrep buffer (Takara Bio, 9182), and the genomic locus of the B2M sgRNA target sequence was amplified by Terra™ PCR Direct Polymerase Mix (Takara Bio, 639270), according to manufacturer instructions (F: GCGCAATCTCCAGTGACAGA, R: GGGATGGGACTCATTCAGGG). The incorporation of indels was determined by Sanger sequencing (GGTGGAAGCTCATTTGGCCAG) of the B2M target sequence amplicon and Tracking of Indels by DEcomposition (TIDE) analysis[27], and B2M loss was phenotypically confirmed by flow cytometry.

## MHC1-TIP with cell lines

Peptide elution from the cells was performed in two variations (Fig. 1B), either in suspension or on-plate. For in-suspension elution, the cells were washed with 1x DPBS once and detached from the culture plate using a 3 min incubation with 0.05% trypsin-EDTA at 37 °C. Detached cells were resuspended in media and washed once with 1x DPBS (GIBCO). From this point, we used protein LoBind tubes (Eppendorf) to reduce sample loss and performed all steps at 4 °C or on ice. We also used LC-grade analytical water to prepare all reagents. The cells were washed once with 150 mM NaCl to remove excess PBS and were then resuspended in 200 μL of mild acid elution buffer. The mild acid elution buffer was adapted from Sturm et al.[4], and consisted of 131 mM citric acid, 66 mM $Na_2HPO_4$, 150 mM NaCl, and 25 mM iodoacetamide (Cytiva), in LC-grade water and adjusted to pH 3.3 with NaOH. After a 3 min incubation in the acid, the samples were briefly centrifuged at $500 \times g$ for 3 min. The supernatant was immediately transferred onto 10 kDa Microcon® Ultracel® filters (MRCPRT010, Merck) that were pre-washed with 150 μL of the mild acid elution buffer at $14,000 \times g$ for 30 min.

After acid elution, the peptides were filtered and desalted in two configurations (Fig. 1B). For the microcentrifuge set up, we utilized the collectors provided with the Microcon® filters (MRCPRT010, Merck) and centrifuged the samples at $14,000 \times g$ for 45–55 min until < 25 μL remained on top of the filters. The filters were washed once more with 100 μL of the mild acid buffer with the same collectors. The filtrate was then desalted using custom 1 mL StageTips, prepared as described previously[6]. Briefly, two punches of a C18 disc (Affinisep SPE-DISKS-BIO-C18-100.47.20) with a blunt-end 14 gauge needle were packed into P1000 pipette tips (Mettler Toledo, 30389279) such that the resin is lodged a few mm above the bottom end of the P1000 tip. The StageTips were activated with 100 μL methanol, washed with 100 μL buffer B [80% acetonitrile, 20% water, 0.1% formic acid (Honeywell)] and equilibrated with 100 μL buffer A (0.1% formic acid in water) twice by centrifuging the StageTips at $1500 \times G$ for 1 min in custom-made 15 mL collection tubes. The ultrafiltered sample was then loaded onto the StageTips and centrifuged at $1000 \times G$ for 5–10 min until all of the sample passed through the C18 resin. The StageTips were then washed once more with buffer A at $1500 \times G$ for 1 min and stored at 4 °C. All centrifugation steps were performed at 4 °C.

For the single-tube configuration of MHC1-TIP, the StageTips were first activated and washed in the same manner as described above. The pre-washed 10 kDa filters (MRCPRT010, Merck) were then fitted on top of the StageTips in 15 mL custom-made collectors that were created by drilling a hole in the caps of 15 mL conical tubes where the StageTips could fit, and the sample was loaded on top of the filters. They were then centrifuged at $4000–5000 \times g$ at 4 °C for 2–3 h until all the sample passed through the filter and C18 resin. The filters were discarded and StageTips were washed once more with buffer A at $1500 \times g$ for 1 min before storage at 4 °C.

For on-plate elution, each well of the 6-well culture plate was washed with 1x PBS twice, followed by 150 mM NaCl twice before 500 μL of mild acid buffer was directly added to each well of the plate. The cells were then incubated in the acid for 3 min on ice, followed by transferring the mild acid from each well onto the pre-washed 10 kDa ultrafilters.

## MHC1-TIP with patient-derived organoid cultures

Tumor resections were obtained from patients with microsatellite instable (MSI) colorectal cancer (CRC) within study NL48824.031.14, approved by the Medical Ethical Committee of the Netherlands Cancer Institute – Antoni van Leeuwenhoek hospital. Written consent was obtained from all patients. All ethical regulations relevant to human research participants were followed. We selected patient-derived organoid (PDO) lines whose MHC-I haplotypes were known:

1. PDO-1: HLA-A*02:01, HLA-A*24:02, HLA-B*15:01, HLA-B*39:01, HLA-B*C03:03, HLA-C*12:03
2. PDO-2: HLA-A*01:01, HLA-A*02:01, HLA-B*07:02, HLA-C*07:02
3. PDO-3: HLA-A*02:01, HLA-A*29:02, HLA-B*13:02, HLA-B*44:03, HLA-C*06:02, HLA-C*16:01.

Organoids were previously derived from tumor resections as described by Dijkstra et al. (2018)[28]. Briefly, tumor tissue derived from needle biopsies was mechanically cut into small pieces and embedded in Geltrex (Geltrex LDEV-free reduced growth factor basement membrane extract, GIBCO). Tumor tissue derived from surgical resections was cut into small pieces, enzymatically digested for 30–60 min with 1.5 mg/mL collagenase II (Sigma-Aldrich), 10 μg/mL hyaluronidase type IV (Sigma-Aldrich) and 10 μM Y-27632 (Sigma-Aldrich) and embedded in Geltrex. After Geltrex solidification for 20 min at 37 °C, cells were overlaid with human CRC organoid medium. Human CRC organoid medium is composed of Ad-DF +++ (Advanced DMEM/F12 (GIBCO) supplemented with 1% Ultraglutamine (Capricorn Scientific), 1% HEPES (GIBCO) and 1% Penicillin/Streptomycin (GIBCO)), 10% Noggin-conditioned medium (producer cell line HEK293-mNoggin-Fc, Lab of Hans Clevers, Hubrecht Institute), 20% R-spondin1-conditioned medium (producer cell line HEK293-T-HA-RSPO1-Fc, Lab of Calvin Kuo, Stanford), 2% B27 supplement without vitamin A (GIBCO), 1,25 mM N-acetylcysteine (Sigma Aldrich), 50 ng/mL human recombinant EGF (BD Biosciences), 500 nM nicotinamide (Sigma Aldrich), 10 μM A-83-01 (Tocris), 10 nM SB202190 (Cayman Chemicals) and 10 nM prostaglandin E2. (Cayman Chemicals).

Cryopreserved organoids were thawed in a 37 °C water bath and immediately dropwise transferred to prewarmed Ad-DF +++ medium. Organoids were embedded in a gel matrix consisting of Ad-DF +++ and Cultrex (RGF BME Type 2, R&D Systems) in a 1:2 ratio and plated dropwise in 6-well plates. After plating, plates were kept inverted at 37 °C for 30 min to allow for polymerization of the gel and 3D growth of the culture. Organoids were subsequently cultured at 37 °C in human CRC organoid medium. Organoids were passaged approximately once a week by incubation in TrypLE Express (GIBCO) for 5–15 min. at 37 °C to dissociate organoids to single cells and replating in fresh Cultrex. After passaging, 10 μM Rock inhibitor Y-27632 (Sigma Aldrich) was added to complete CRC medium for the first 2–3 days.

For immunopeptidome profiling, three wells of a 6-well plate per line were treated with IFNy (Peprotech) for 24 h. The organoids were cultured in their complete CRC medium with the addition of 200 ng/mL IFNy. The organoids were harvested by vigorous pipetting with cold PBS and incubated on ice for 15 min to dissolve the gel matrix. The organoids were washed with cold PBS twice and once with 150 mM NaCl. The cells were then resuspended in 200 μL mild acid elution buffer, incubated on ice for 3 min and spun down at $300 \times g$ for 4 min at 4 °C. The supernatant was then processed with the microcentrifuge workflow of MHC1-TIP as described above.

## MHC1-TIP with patient-derived tumour fragments

This study uses a tumor sample from a patient diagnosed with renal cell carcinoma (RE) who underwent surgical resection at the Netherlands Cancer Institute (NKI-AVL), The Netherlands. The institutional review board of the NKI-AVL approved the study (CFMPB484), which was conducted in accordance with the ethical guidelines. All patients in the study consented to research usage of material not required for diagnostic use via written informed consent. All ethical regulations relevant to human research participants were followed.

Fresh tumor tissue was collected on ice in collection medium [Roswell Park Memorial Institute (RPMI) 1640 medium (Thermo Fisher Scientific) supplemented with 2.5% fetal bovine serum (FBS) (Sigma-Aldrich) and 1% penicillin-streptomycin (Roche)] and directly processed into 3D pieces of 1–2 $mm^3$. To account for intratumoral heterogeneity, patient-derived tumor fragments (PDTFs) were pooled from different regions, suspended in 1 mL of FBS containing 10% dimethylsulfoxide (DMSO; Sigma-Aldrich), and cryopreserved in liquid nitrogen until use.

PDTF cultures were performed as previously described in refs. 10,29. Briefly, cryopreserved PDTFs were thawed in a water bath until a small ice cube remained, then thoroughly washed with a wash medium (DMEM

containing 10% FBS and 1% penicillin-streptomycin). Individual PDTFs were embedded in an artificial extracellular matrix {1.1% sodium bicarbonate (Sigma-Aldrich), 1 mg/mL rat-tail collagen I (Corning), 2 mg/mL Matrigel (Cultrex UltiMatrix, reduced growth factor basement membrane extract, R&D Systems), tumor medium [DMEM supplemented with 1 mM sodium pyruvate (Sigma-Aldrich), 1× MEM non-essential AA (Sigma-Aldrich), 2 mM L-glutamine (Thermo Fisher Scientific), 10% FBS, and 1% penicillin-streptomycin]}. One PDTF was seeded per well in a flat bottom 96-well plate, and additional tumor medium was added. Cultures were incubated at 37 °C. After 24 h, each PDTF was transferred into cold PBS and washed once more with cold PBS in 5 mL tubes (Eppendorf), followed by washing with 150 mM NaCl. The liquid surrounding the fragment was pipetted and discarded. The fragments were then incubated on ice for 5 min in 200 µL mild acid elution buffer. The acid buffer was then pipetted off from around the fragment and transferred onto pre-washed 10 kDa filters in the microcentrifuge setup on MHC-1TIP.

## Proteomics

Cell pellets, organoids and tumour fragments were either immediately processed for proteomics after mild acid elution, or snap frozen in a dry ice and ethanol bath and stored at −20 °C until further processing. The pellets were kept on ice until lysis or snap-freezing. The 'Sample Preparation by Easy Extraction and Digestion (SPEED)' method was used for proteomics sample preparation[30]. Briefly, the samples were lysed with trifluoroacetic acid (TFA) (Uvasol® for spectroscopy, Merck) at a sample/TFA ratio of 1:4 (v/v). Cell lines and organoids were incubated in TFA at room temperature (cell lines and organoids for 2 min and tumour fragments for 20 min) until the pellet fully dissolved. TrisBase 2 M was then added to all samples at 10x the volume of TFA used. Protein concentrations were then estimated using a NanoDrop 2000c spectrophotometer (Thermo Scientific) with the Protein A280 method. The entire protein lysate was then incubated at 95 °C for 5 min after spiking-in Tris(2-carboxyethyl)phosphine (TCEP) to a final concentration of 10 mM and 2-Chloroacetamide (CAA) to a final concentration of 40 mM. The volume of lysate containing 10–20 µg of protein was then diluted in 50 or 100 µL of dilution buffer (1:10 of TFA/TrisBase, 10 mM TCEP and 40 mM CAA) before digestion with Trypsin (V5111, Promega) for 20 h at 37 °C, at a Trypsin:protein ratio of 1:100 µg. at a protein/enzyme ratio of 50:1. After digestion, the samples were acidified with 8% TFA and desalted using StageTips[6].

## Immunoprecipitation-based immunopeptidomics

Cell pellets were resuspended in 500 µL of lysis buffer composed of 0.25% sodium deoxycholate, 0.2 mM iodoacetamide, 1 mM EDTA, 1% octyl-β-d-glucopyranoside (O3757-5 mL, Sigma), 1x cOmplete Protease inhibitor (11836145001, Roche) and 0.1 mM phenylmethylsulfonyl fluoride (93482-250mL-F, Roche)] in protein LoBind tubes and incubated on ice for 1 hour. The lysates were then cleared by centrifugation at maximum speed for 45–60 min at 4 °C. To the supernatant, 40 µL of protein A sepharose beads (10-1041, Thermo Fisher Scientific) pre-washed with the lysis buffer were added along with 50 µg of the W6/32 antibody. Following an overnight incubation on a rotor at 4 °C, the supernatant was removed and the beads-antibody-peptide-MHC complexes were washed 4 times with 200 µL of 150 mM NaCl in 20 mM Tris-HCl (pH 8.0), twice with 400 mM NaCl in 20 mM Tris-HCl (pH 8.0), twice with 150 mM NaCl in 20 mM Tris-HCl (pH 8.0) and 4 more times with 20 mM Tris-HCl (pH 8.0). The washes were carried out by centrifugation at 900 RPM for 2 min at 4 °C. To elute peptides from the beads and MHC-I complexes, the beads were resuspended in 100 µL of 1% TFA. The eluted peptides in 1% TFA were then desalted with 1 mL StageTips[6] as described above.

## Mass spectrometry

Before mass spectrometry, peptides were eluted from StageTips into protein LoBind tubes using 30 µL of buffer B (80% acetonitrile, 20% LC-grade water, 0.01% formic acid), followed by drying in a vacuum centrifuge at 45 °C. Immunopeptidomes were then resuspended in 7.5 µL buffer A and half of

the sample was used for measurement on the mass spectrometer. Proteomes were resuspended in buffer A (0.01% formic acid in LC-grade water) at a concentration of 1 µg/µL and 0.5 to 1 µg was injected into the LC.

Samples were analyzed by single-shot LC-MS/MS on the Orbitrap Astral mass spectrometer (Thermo Scientific) connected to a Vanquish Neo nano-LC system (Thermo Scientific) using a 30 samples-per-day (30SPD) LC-MS method. The Vanquish Neo was operated in trap-and-elute mode with peptides loaded onto a Pepmap 100 C18 5 µm trap column (300 µm x 5 mm, Thermo Scientific) before separation on the analytical column. For proteomes, the analytical column was either ES906 (2 µm/ 150 µm x 150 mm, Thermo Scientific) or AUR3-25075C18-TS (1.7 µm/ 75 µm x 25 cm, IonOpticks AU), whereas MHC1-TIP immunopeptidomes were separated on either AUR3-25075C18-TS or AUR3-15075C18-TS (1.7 µm/75 µm x 15 cm, IonOpticks). All columns were heated at 50 °C and mounted onto an Easyspray ion source (Thermo Scientific) with ion spray voltage set to 1500 V or higher. Solvent A was 0.1% formic acid/water and solvent B was 0.1% formic acid/80% acetonitrile in water and peptides were separated at flow rates of 0.4 µL/min (AUR3-25075C18-TS and AUR3-15075C18-TS) or 0.8 µL/min (ES906) in a circa 36-min effective gradient containing a non-linear increase from 8% to 45% solvent B, followed by a wash-out at 99% solvent B and equilibration at the end using the "fast equilibration" script in combined control mode with a 1450 bar pressure limit. For proteomes, the Orbitrap Astral was run in DIA mode, with full MS scans being collected in the Orbitrap analyzer at 240,000 resolution at m/z 200 over a 380–980 m/z range. The normalized AGC target was set to 500% (equivalent to 5e6 charges) and the maximum injection time was 5 ms. For DIA MS2, a normalized HCD collision energy of 25% was applied to a 380-980 m/z precursor range using non-overlapping isolation windows of 2Th, with window placement optimization turned on. Scans were acquired in the Astral analyzer over a 100–1000 m/z range, with the normalized AGC target set to 500% (equivalent to 5e4 charges) and a maximum injection time of 3 ms. Immunopeptidomes were measured using either the DIA settings described above, or using a DDA method with full MS resolution in the Orbitrap set to 240,000 and peptide precursors with charge states 1–6 being sampled for MS/MS in 0.6 s. cycles. MS1 mass range was 400–1500 m/z, the normalized AGC target was 500% and maximum IT was set to 40 ms. For MS2, the intensity threshold was $5 \times 10^3$ and a 20 sec. exclusion duration was used. Precursors isolated in the quadrupole within a 1.2 m/z isolation window were fragmented with a normalized HCD collision energy of 25% and MS2 spectra were acquired in the Astral analyzer with scan range 120–1800 m/z, 200% normalized AGC target and a 30 ms maximum injection time.

For DIA spectral library generation, DDA immunopeptidomics data was created using our default LC-MS/MS setup for immunopeptidomics. Immunoprecipitated peptides were reconstituted in 0.1% formic acid and analyzed on an Orbitrap Exploris 480 Mass spectrometer connected to an Evosep One LC system (Evosep Biotechnology, Odense, Denmark). Peptides were eluted from Evotip Pure™ (Evosep) tips directly on-column and separated using the pre-programmed "Extended Method" (88 min gradient) on an EV1137 (Evosep) column with an EV1086 (Evosep) emitter. Nanospray was achieved using the Easy-Spray NG Ion Source (Thermo Scientific). On the Exploris 480, data-dependent acquisition was performed as follows. Full scan MS was acquired at resolution 60,000 in the orbitrap with MS1 mass range 350-1700 m/z, normalized AGC target was set to 100% and maximum injection time was 50 ms. Dynamic exclusion was set to 10 s. and MS2 spectra were acquired at 15,000 resolution. The top 10 precursors per cycle were HCD fragmented when their charge states were 2–4, whereas the top 5 precursors per cycle were subjected to HCD fragmentation if they were singly charged. MS2 isolation window was 1.1 m/z, the normalized collision energy was 30, the normalized AGC target was set to 50% and the maximum injection time was 100 ms.

## Data analysis

### Analysis of DDA immunopeptidomes.
DDA raw files from MHC1-TIP experiments were analyzed using FragPipe (v22 or v23), using the built-in

'nonspecific-HLA-C57' workflow with all preset parameters. MS data was searched using MSFragger[14]. The initial precursor and fragment mass tolerances were set to 20 ppm. Spectrum deisotoping, mass calibration, and parameter optimization were enabled. The isotope error was set to "0/1". The reviewed Homo sapiens protein sequence database obtained from UniProt (downloaded on 18 December 2024), appended with common contaminants and decoys, was used in the search. The enzyme cleavage was set to "nonspecific" and the peptide length was restricted to 7–25. Carbamidomethylation of cysteine was specified as a fixed modification. Cysteinylation, oxidation of methionine, N-terminal acetylation, and Pyro-glu from E and Q were set as variable modifications. The maximum number of variable modifications for each peptide was set to 3. MSBooster and Percolator were used to predict the RT and MS/MS spectra, and to rescore PSMs[15]. FDR filtering was performed using Philosopher[31]. IonQuant[32] was used to extract peptide ion intensities for all PSMs.

**Analysis of DIA immunopeptidomes with a DDA-based spectral library using FragPipe and DIA-NN.** For cell line experiments presented in this paper (Figs. 1E–G, and 2B–D), quantification from DIA raw files was extracted using a spectral library generated from DDA data. Immunopeptides were obtained from three replicates of $10^8$ A375 cells each using the immunoprecipitation method and were measured in DDA mode on the Orbitrap Exploris. The spectral library was generated from these DDA files using FragPipe (v21) using the built-in 'nonspecific-HLA-C57' workflow, as described above. The final FDR-filtered PSMs and the raw DDA spectral files were used by EasyPQP to generate the spectral library, filtered to 1% peptide-level false discovery rate (FDR). This DDA-based library.tsv file built by FragPipe was then passed to standalone DIA-NN (v2.0)[33,34] for extraction of quantification from DIA data, along with the human proteome reference fasta file (UP000005640) for library annotation. Mass accuracy was set to 10.0 and MS1 accuracy to 4.0. To determine the appropriate scan window, representative raw files from the experiment were first analyzed as 'Unrelated runs' and the DIA-NN suggested scan window radius was used for a subsequent full analysis. Match between runs (MBR) was enabled for all experiments, with the exception of the comparison between MHC-1TIP and IP (Fig. 1E-F).

**Analysis of Astral DIA immunopeptidomes using MSFragger-DIA and FragPipe.** For direct DIA data analysis of the patient-derived tumour fragments, we used the 'Nonspecific-HLA-DIA-Astral' workflow in FragPipe with all built-in settings. Astral narrow-window DIA MS data was searched using MSFragger-DIA[13]. The initial precursor and fragment mass tolerances were set to 10 and 20 ppm, respectively. Spectrum deisotoping, mass calibration, and parameter optimization were enabled. The same reviewed protein sequence database was used as above. The enzyme cleavage was set to "nonspecific" and the peptide length was restricted to 7–15. Carbamidomethylation of cysteine was specified as a fixed modification. Cysteinylation, oxidation of methionine, and N-terminal acetylation were set as variable modifications. To ensure the highest quality of the resulting spectral library, MSFragger-DIA options were set to filter out PSMs with no precursor ion feature detected in the MS1 data or having less than 5 fragments matched in a single MS/MS scan. The rest of the analysis (rescoring with MSBooster and Percolator, FDR filtering with Philosopher, and spectral library building with EasyPQP) was performed as described above. The resulting library.tsv file built by FragPipe was used to extract quantification from the same DIA files using DIA-NN version 1.8.2 integrated in the FragPipe computational platform.

**Proteomics data analysis.** Proteomics data was analyzed with DIA-NN as described above, without the immunopeptidome spectral library. An in-silico predicted human proteome spectral library instead was first generated using the UniProt database in DIA-NN (downloaded on 03 January 2025), which was then subsequently used to search raw files.

**Downstream analyses and statistics.** Matched proteomes and immunopeptidomes from cell lines and organoids were obtained from three distinct replicates processed in parallel for statistics testing, with the exception of one untreated proteome sample from the TGFβ perturbation experiment which was excluded due to a technical error resulting in no proteins identified in the sample.

The output tables generated by FragPipe (DDA immunopeptidomes), DIA-NN using FragPipe built libraries (for DIA immunopeptidome data), or DIA-NN library-free mode (proteome data), were further processed and analysed in R. Filtering for strong (predicted affinity <= 100 nM) and weak binders (predicted affinity between 100 and 500 nM) was performed using MHCFlurry[7] for MHC class I (with peptides of lengths 7–16 as per MHCflurry requirements) and NetMHCIIPan 4.3 for MHC class II (peptides of length 9 and longer according to NetMHCIIPan requirements). MHC-I haplotypes for the A375 cell line and JY cells were obtained from previously published data, through the Cellosaurus database. Immunopeptidomes from patient-derived tumour fragments were filtered for length 9 and a match to the MHC-I characteristic motif using GibbsCluster[9] before quantitative analyses.

Differential protein expression and differential antigen enrichment analyses were carried out using DEP[35] from the DIA-NN protein group matrix output file or the DIA-NN precursor matrix output file, respectively. For immunopeptidomics data, intensities from different versions of the same precursor, such as with different modifications or charges were summed up. Immunopeptidomics data from A375 cells treated with TGFβ, and the patient-derived tumour fragments were imputed using a mixed approach wherein a peptide that is missing in all replicates of a group was imputed by a probabilistic minimal value approach ('MinProb') while missing values occurring at random were imputed with the k-nearest neighbours ('knn') method implemented in the imputation function of the DEP R package. Proteomics data from A375 cells treated with mild acid, and TGFβ were imputed with the k-nearest neighbours ('knn') method alone. Differential abundance testing was performed by applying empirical bayes statistics using limma[36], and the moderated t-statistics were used to compute FDR-corrected p-values using the Benjamini Hochberg method, also implemented in the DEP R package. For absolute protein copy numbers, proteomic ruler[16] method was applied in R, using MS1 intensities in the report file from DIA-NN, filtered to include only precursors with a global protein group q-value < 0.01. Heatmaps and hierarchical clustering were generated using the 'pheatmap' package. Gene set enrichment analyses were carried out using 'fgsea'[37] with Hallmark gene sets downloaded from MSigDB. Proteins were ranked by their p-values adjusted for the direction of fold change.

For the tumour fragment immunopeptidomes, intensities from the precursor matrix were normalized by calculating the fraction that a peptide contributes to the summed total intensity of all peptides quantified in that sample. The fractions were further scaled by B2M copy numbers per cell computed using the 'proteomic ruler' method.

### Statistics and reproducibility

For all figures, where applicable, individual data points are displayed and sample sizes are indicated in the figure legends. Replicates were defined as distinct samples processed in parallel. Statistical analyses were performed in R to assess multimodal and quantitative data generated by the MHC1-TIP workflow. Spearman correlation analyses between log-transformed peptide intensities were conducted using the 'stat_cor' function implemented in 'ggpubr'. Distributions of peptide numbers across sample groups were visualized using boxplots generated by the 'ggplot2' package. The boxes show the interquartile range wherein the central line represents the medians and the whiskers are extreme values upon removing outliers. Differential protein and peptide enrichment analyses from the proteomics and immunopeptidomics data were performed using the 'DEP' R package, which included normalization, variance stabilization, and missing value imputation followed by empirical Bayes statistics, and FDR correction, described above in detail.

Significance threshold was set to FDR-adjusted $p$-value < 0.05 and $\log_2$ fold changes > 1 to define significance.

## Reporting summary

Further information on research design is available in the Nature Portfolio Reporting Summary linked to this article.

## Data availability

The data presented in this study, including mass spectrometry raw data and output tables are available through the ProteomeXchange Consortium via the PRIDE partner repository under accession number PXD065279, and through our GitHub repository at https://github.com/LindeboomLab/MHC1-TIP.

## Code availability

Custom code used for data analysis is available in our GitHub repository[38] at https://github.com/LindeboomLab/MHC1-TIP.

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

## Acknowledgements

K.K.D. is supported by an NWO/ZonMW Veni fellowship under project number 09150162210100 and a KWF Young Investigator Grant under project number 2025-EXPL/16959.; R.A. is supported by the Dutch Cancer Society (KWF-13647), the European Research Council (Horizon ERC-2023-ADG-101141245), and the AvL Foundation. J.C. is supported by the Dutch Cancer Society (KWF-15986). Work in the Lindeboom lab is supported by the Dutch Cancer Society (KWF-16496), the Cancer Research Institute (CRI5377), the Dutch Research Council (NWO Open Competitie ENW - M OCENW.M.22.351), the European Research Council (Horizon ERC-2024-StG-101162742), and the AvL Foundation. Research at the Netherlands Cancer Institute is supported by institutional grants of the Dutch Cancer Society and of the Dutch Ministry of Health, Welfare and Sport. Work in the Nesvizhskii lab (A.I.N, F.Y.) was funded in part by National Institutes of Health (NIH) grants R01-GM-09423 and U24-CA271037 (to A.I.N.).

## Author contributions

R.G.H.L. conceived the study. M.B. and R.G.H.L. designed the methodology and analyses. M.B. performed the data analysis and made figures. M.B., D.B., F.D.J., R.C.P.J., V.D.A., M.C.K., J.R.M., L.G., J.C., O.B. performed experiments. F.Y. and A.I.N. provided software and input for direct DIA analyses. M.B. and R.G.H.L. wrote the manuscript. M.B., R.G.H.L., F.D.J., R.C.P.J., M.C.K., J.R.M., R.A, K.K.D., D.S.T. and A.I.N. edited the manuscript.

## Competing interests

A.I.N. is the Founder of Fragmatics and serves on the scientific advisory boards of Protai Bio and Infinitopes. F.Y. is a paid consultant for Fragmatics. A.I.N. and F.Y. have a financial interest due to the licensing of MSFragger and IonQuant to commercial entities. Other authors declare no competing interests.
