## [Transparent Peer Review file · Communications Biology]

MHC1-TIP enables single-tube multimodal immunopeptidome profiling and uncovers intratumoral heterogeneity in antigen presentation

Corresponding Author: Dr Rik Lindeboom

Version 0:

Reviewer comments:

Reviewer #1

(Remarks to the Author)

The authors present a study where they developed a one pot method for HLA peptide isolation using mild acid elution. The advantages of this method include the reduction in starting material and the enrichment of HLA-I peptides that were likely presented on the cell surface in comparison to the conventional immunopurification protocols that require more input material and may contain HLA complexes on their way to the surface of the cell for display. The disadvantage of this protocol is that it requires the use of viable cells or material that is expanded from primary tissue samples. The authors call this primary patient tissues, and instead, it would be more accurate to use "patient samples derived from an ex vivo tumor fragment method".

In terms of novelty, a similar approach was recently described by Beyrle et al (doi: <https://doi.org/10.1101/2024.12.20.628848>). A novel aspect of this manuscript is the application of mild acid elution to samples generated using an ex vivo tumor fragment method. The use of B2M measured in the matched proteome as a proteomic ruler is also a novelty.

The authors were very thoughtful in their experimental design. I liked the use of a B2M KO as a negative control to look at non-specific binding peptides. I also thought the use of IFN treated organoids was a nice application to show how multiomics pairs with this mild acid elution protocol to demonstrate these data recapitulate known biology and can be used to look at perturbations of organoids.

The one major flaw is that the authors did not discuss how acid elution protocols and analysis are impacted when analyzing cells that express both HLA-I and HLA-II. For example, A375 cells express both HLA-I and HLA-II, so it is expected that an acid elution protocol would result in a mixture of HLA-I and HLA-II peptides. Based on the methods, it appears that the length distributions do not reflect the presence of HLA-I and HLA-II peptides because the DIA analysis is length filtered to 7-15AA. In Figure 2, the authors also show, as expected, that HLA-DR increased with IFN, so the IFN samples should also have more HLA-II peptide contamination if it is expressed. Do any of these PDO also have HLA-II, as the heat maps from the multiomic data suggest they may. If so, those should also be reanalyzed and deconvoluted to subset the HLA-I and HLA-II peptides from the dataset.

I believe the authors should go back and evaluate how much HLA-II peptides are possibly in their data by opening the length filters up to at least 25 amino acids. If both HLA-I and HLA-II are present in the data, length filtering and HLA binding prediction are two methods that can be used to deconvolute the data. There may also be instrument method parameters (such as charge state filtering) and other analysis parameters that impact the ability to identify HLA-I and HLA-II peptides that should be discussed as well. If the authors can do this, their one-pot protocol gets much more interesting because a claim can be made that HLA-I and HLA-II data can be isolated together and deconvoluted to reduce the sample processing time even further, as serial protocols take double the amount of time. Discussing both HLA-I and HLA-II co-isolations would further differentiate this study from the previous ones and increase the novelty.

I expect the HLA-II predictions to not be as robust as HLA-I, and this should be discussed as a limitation of deconvolution. It may also be the case that the instrument parameters used for HLA-I are not optimized for HLA-II, so if you want to identify

both well, instrument methods would need further optimization. This would make for a nice discussion point in the manuscript.

Some additional comments to improve the clarity of the manuscript:

- HLA-II analysis on all samples. I expect this may result in an additional figure
- Clearly name if DDA or DIA is used in the figure legends
- Please report the percentage of HLA-I and HLA-II peptides that are predicted to bind to the alleles present in these cells, as the global motifs do not enable this assessment. For example, HLA peptide isolations using antibodies can have ~90-95% of peptides assigned to alleles, while acid elution often has ~80%, or less assigned. These are important metrics to show and discuss.
- Figure 1 B does not convey the material/strategy used for HLA isolation. Is it hydrophobicity based or size exclusion based? Adding these labels to the schematic would be helpful.
- The Methods are not detailed enough for readers to reproduce this protocol. For example, which Microcon filters were used and what type of StageTip was used? Having these labels in Figure 1 schematics would help readers follow your protocol and reproduce it. In addition, if the filters and StageTips were communicably purchased, please include part numbers and if the stage tips were made in-house, how much material was used? Please do not make readers open another reference to find key details to reproduce a protocol.
- HLA-II typing will need to be reported for all samples. Please include full typing that includes HLA-DRB3/4/5 if present.
- The claim that MAE only obtains HLA peptides from the cell surface is not fully supported. For example, soluble HLA can be released into the media by membrane-bound metalloproteinases, and this data could include sHLA peptides. Although in theory MAE should be enriched with HLA peptides from the cell surface, it has not been fully quantified how much more enriched it is compared to HLA peptide isolations using w6/32. Unless the authors do a direct comparison (HLA peptide immunopurification vs. MAE), and can show an orthogonal method, such as intracellular staining with w6/32 to quantify the amount of HLA peptide complexes that could be coming from inside the cells in immunopurification datasets, than this claim should be removed or toned down.

Reviewer #2

(Remarks to the Author)

Thank you so much for giving me the opportunity to review the manuscript, "MHC1-TIP enables single-tube multimodal immunopeptidome profiling and uncovers intratumoral heterogeneity in antigen presentation" by Lindenboom et al. The authors developed a new sample preparation system called "MHC1-TIP", improved version of mild acid elution method, that enables immunopeptidomics and proteomics simultaneously from the identical samples. This helped the detailed comparison of abundance of source proteins as well as the epitope presentation with higher resolution than separately-prepared previously reported approaches. The sample preparation sounds reasonable, while the efficacy of the immunopeptidomics by MHC1-TIP than previous & general IP method may not be fully described. To improve the impact, a few suggestions are mentioned below.

Major Comments

1. Since the authors mentioned, "Previous implementation of mild acid elution has possibly been hampered by coelution of contaminant biomolecules that interfere with the identification of relevant antigens...", the assessment of the "purity" of identified peptides by MHC1-TIP would better describe the advantage of MHC1-TIP. Especially in Figure 1F, the number of peptides identified by IP-method seems, at least to me, not that different when compared to the MHC1-TIP at the number of 100,000 cells. Therefore, it may be better to show how much of the identified peptides are the predicted binders to the sample-unique HLA allotype. One suggestion to extract the predicted binders, the NetMHCpan (<https://services.healthtech.dtu.dk/services/NetMHCpan-4.1/>) can be used. Any algorithms will do (it depends author's choice) to predict the HLA-matched binders, while it should become informative to assess the "purity" (HLA-matched immunopeptides) of MS-identified to describe the advantage of MHC1-TIP.

2. Unlike the cell lines and the organoid samples (in-vitro cultures), the tissue samples are expected to include physiological peptides that can be derived from peripheral blood. This is another concern about the identified peptide by MHC1-TIP in relation to the major comment 1, Was there any difference in the purity of identified (immuno)peptides between MHC1-TIP and IP-methods? The comparison of cell line/ organoid vs tissue, as well as the IP vs MHC1-TIP, from the perspective of the purity of immunopeptides (ratio of predicted binders/ identified peptides) should be validated to describe the advantage of MHC1-TIP. This would be more informative to the readers/ researchers who want to try the MHC1-TIP with clinical tissue samples.

Minor Comments

3. The negative control of B2M knock-out for FACS studies seemed a bit inconsistent. In Figure 1A, the B2M knock-out as negative control exhibited the fluorescent intensity of 1×10^3 , while the negative control described in Supplementary figure 1B exhibited the fluorescent intensity of almost 1×10^2 . In supplementary figure 1C, the distribution of B2M knock-out signal by FACS became rather broader. It may be better to confirm and show the peak pattern of exact genomic sequence for the

clonality. If possible, it is recommended to describe the mutation accordingly to the HGVS Nomenclature (<https://hgvs-nomenclature.org/stable/recommendations/general/>) to show what kind of mutation was induced by CRISPR in B2M gene. In addition, the antibody of isotype control for W6/32 (IgG2a, if I remember right?) should be useful to distinguish the background and autofluorescence noise from the genuine signal of HLA when conduct the FACS studies.

4. Have the authors ever tried to identify the neoepitopes in A375? I presume that in the context of "Previous implementation of mild acid elution has possibly been hampered by coelution of contaminant biomolecules that interfere with the identification of relevant antigens...", the word "relevant antigens" mean the neoepitopes. It would be nice to know whether MHC1-TIP can produce the sample that includes neoepitope.

Version 1:

Reviewer comments:

Reviewer #1

(Remarks to the Author)

The authors have addressed a majority of my concerns, so in my view the revised manuscript is suitable for publication.

A minor suggestion that could be incorporated during the proofing stage is that there is inconsistent use of HLA-I vs. MHC1 throughout the manuscript. It may be beneficial to stick with one naming convention throughout.

Reviewer #2

(Remarks to the Author)

The author carefully addressed most of the comments and suggestions in the revised manuscript, and these issues have now been satisfactorily resolved.

Regarding the feasibility of finding neoepitopes, since A375 is listed in the CCLE (COSMIC) list, so from my understanding, it is possible to search for mutation-carrying neoepitopes in A375. However, since this is not the main argument of this manuscript, I think it is acceptable to limit the discussion to tumor-associated antigens.

As for the methodology devised by the authors, while purity issues remain, this also depends on the end-user's objectives and preferences (whether they prefer Class 1 and Class 2 separately or if combining them is acceptable). Hence, this response is also considered acceptable.

Based on this review process, I believe that the revised manuscript now provides content satisfactory to readers.

Reviewer #1 (Remarks to the Author):

The authors present a study where they developed a one pot method for HLA peptide isolation using mild acid elution. The advantages of this method include the reduction in starting material and the enrichment of HLA-I peptides that were likely presented on the cell surface in comparison to the conventional immunopurification protocols that require more input material and may contain HLA complexes on their way to the surface of the cell for display. The disadvantage of this protocol is that it requires the use of viable cells or material that is expanded from primary tissue samples. The authors call this primary patient tissues, and instead, it would be more accurate to use “patient samples derived from an ex vivo tumor fragment method”.

We thank the reviewer for their suggestion. We have revised the manuscript accordingly. The term “primary patient tissues” has been replaced with “patient samples derived from an ex vivo tumor fragment method” throughout the text to more accurately describe the source of the material.

In terms of novelty, a similar approach was recently described by Beyrle et al (doi: <https://doi.org/10.1101/2024.12.20.628848>). A novel aspect of this manuscript is the application of mild acid elution to samples generated using an ex vivo tumor fragment method. The use of B2M measured in the matched proteome as a proteomic ruler is also a novelty.

The authors were very thoughtful in their experimental design. I liked the use of a B2M KO as a negative control to look at non-specific binding peptides. I also thought the use of IFN treated organoids was a nice application to show how multiomics pairs with this mild acid elution protocol to demonstrate these data recapitulate known biology and can be used to look at perturbations of organoids.

The one major flaw is that the authors did not discuss how acid elution protocols and analysis are impacted when analyzing cells that express both HLA-I and HLA-II. For example, A375 cells express both HLA-I and HLA-II, so it is expected that an acid elution protocol would result in a mixture of HLA-I and HLA-II peptides. Based on the methods, it appears that the length distributions do not reflect the presence of HLA-I and HLA-II peptides because the DIA analysis is length filtered to 7-15AA. In Figure 2, the authors also show, as expected, that HLA-DR increased with IFN, so the IFN samples should also have more HLA-II peptide contamination if it is expressed. Do any of these PDO also have HLA-II, as the heat maps from the multiomic data suggest they may. If so, those should also be reanalyzed and deconvoluted to subset the HLA-I and HLA-II peptides from the dataset.

I believe the authors should go back and evaluate how much HLA-II peptides are possibly in their data by opening the length filters up to at least 25 amino acids. If both HLA-I and HLA-II are present in the data, length filtering and HLA binding prediction are two methods that can be used to deconvolute the data. There may also be instrument method parameters (such as charge state filtering) and other analysis parameters that impact the ability to identify HLA-I and HLA-II peptides that should be discussed as well. If the authors can do this, their one-pot protocol gets much more interesting because a claim can be made that HLA-I and HLA-II data can be isolated together and deconvoluted to reduce the sample processing time even further, as serial protocols take double the amount of time. Discussing both HLA-I and HLA-II co-isolations would further differentiate this study from the previous ones and increase the novelty.

We thank the reviewer for their compliments and for highlighting this great suggestion to search for possible HLA-II peptide co-elution in our method. Here, it is important to note that the peptide length filter of 7-15 was only applied for direct DIA analyses with the patient-derived tumour fragments and A375 cell line data shown in Supplementary Figure 3C-D. In all the other analyses, we used a 7-25 amino acid length filter, which should not select against class II immunopeptides (see for example a typical MHC class II immunopeptidome in Reviewer Figure 1). Specifically, figures 1C-D and 1H-I show DDA data where peptide lengths were set to 7-25, and figures 1E and 1F display DIA datasets that were analyzed using a spectral library generated with a length filter of 7–25 residues.

To investigate if MHC1-TIP data also includes class II antigens, we reanalyzed the unbiased DDA search (up to 25 amino acids long) in Figure 1D, which does not reveal an additional elution of longer peptides beyond what is expected from class I peptides. Inspired by the suggestion by the reviewer to examine the expected increase in HLA-II peptides, we reanalyzed experiments shown in Figures 1G and searched for MHC class II binders using NetMHCIIpan 4.3. In IFN γ treated samples, we only identified 2-6 peptides with a predicted affinity to MHC class II of ≤ 500 nM. As melanoma cells might not be the best antigen-presenting cells for HLA-II peptides, and as the PDOs did not display any HLA-II expression by flow cytometry, we applied our method with a 'JY' B-cell line that is known to have high amounts of HLA-II antigen presentation. Using the same workflow, we observed only 4 strong MHC class II binders (predicted binding affinity ≤ 100 nM) and 39 weak binders (predicted binding affinity between 100 nM and 500 nM). This indicates that MHC1-TIP does not suffer from co-elution of MHC class II antigens and that our method is specific to MHC class I antigens, which is in line with earlier biochemical evidence by Sugawara et al., (*Journal of Immunological Methods*, 1987), that acid elution functions through the dissociation of B2M in a MHC class I specific manner. We have included this important point about HLA specificity and the absence of co-elution in supplementary figure 1, and discussed it in the main text (lines 195-206) of the revised manuscript.

I expect the HLA-II predictions to not be as robust as HLA-I, and this should be discussed as a limitation of deconvolution. It may also be the case that the instrument parameters used for HLA-I are not optimized for HLA-II, so if you want to identify both well, instrument methods would need further optimization. This would make for a nice discussion point in the manuscript.

We thank the reviewer for this insightful comment. Our DDA method used to measure the MHC1-TIP samples included charge states 1-6. Therefore, we believe that peptide lengths typically associated with HLA-II presentation (12–16 amino acids) would be well within the range detectable by our current settings, and we would not expect significant bias against such peptides based on length alone. Using the same chromatography and mass spectrometry set-up that we used for MHC1-TIP samples, we routinely measure immunopurification-based HLA-II peptidomes, further underscoring that our set-up is technically capable of detecting HLA-II peptides. We show below an HLA-II immunopeptidome obtained from A375 cells using the pan-HLA-II IVA-12 antibody with the same LC-MS DDA setup reported in our manuscript (Reviewer Figure 1).

Reviewer Figure 1: Typical MHC class II immunopeptidome and its peptide length distribution generated with the same LC-MS/MS setup as used in MHC1-TIP using an anti-HLA-DR antibody pull down, highlighting our ability to detect HLA-II antigens.

Some additional comments to improve the clarity of the manuscript:

- HLA-II analysis on all samples. I expect this may result in an additional figure

We have included the above described HLA-II analyses on the A375 and JY cells in Supplementary Figure 1 of the revised manuscript.

- Clearly name if DDA or DIA is used in the figure legends

We have added this information to all figure legends now for improved clarity.

- Please report the percentage of HLA-I and HLA-II peptides that are predicted to bind to the alleles present in these cells, as the global motifs do not enable this assessment. For example, HLA peptide isolations using antibodies can have ~90-95% of peptides assigned to alleles, while acid elution often has ~80%, or less assigned. These are important metrics to show and discuss.

This information is now displayed in Supplementary Figure 1 and discussed below.

- Figure 1 B does not convey the material/strategy used for HLA isolation. Is it hydrophobicity based or size exclusion based? Adding these labels to the schematic would be helpful.

After acid elution, the peptides are filtered by size exclusion (< 10 kDa) and captured on a C18 resin for desalting and concentration. We have added this to the figure legend for better clarity.

- The Methods are not detailed enough for readers to reproduce this protocol. For example, which Microcon filters were used and what type of StageTip was used? Having these labels in Figure 1 schematics would help readers follow your protocol and reproduce it. In addition, if the filters and

StageTips were communicably purchased, please include part numbers and if the stage tips were made in-house, how much material was used? Please do not make readers open another reference to find key details to reproduce a protocol.

We have added this information to the methods section in more detail (lines 544-577 of the revised manuscript).

- HLA-II typing will need to be reported for all samples. Please include full typing that includes HLA-DRB3/4/5 if present.

We appreciate the reviewer's suggestion to include full HLA-II typing. However, given our new insights into the absence of HLA-II peptide co-elution in MHC1-TIP shown in the comments above, we believe that full HLA-typing is not necessary anymore.

- The claim that MAE only obtains HLA peptides from the cell surface is not fully supported. For example, soluble HLA can be released into the media by membrane-bound metalloproteinases, and this data could include sHLA peptides. Although in theory MAE should be enriched with HLA peptides from the cell surface, it has not been fully quantified how much more enriched it is compared to HLA peptide isolations using w6/32. Unless the authors do a direct comparison (HLA peptide immunopurification vs. MAE), and can show an orthogonal method, such as intracellular staining with w6/32 to quantify the amount of HLA peptide complexes that could be coming from inside the cells in immunopurification datasets, than this claim should be removed or toned down.

This is a good point. We have removed this claim from our manuscript.

Reviewer #2 (Remarks to the Author):

Thank you so much for giving me the opportunity to review the manuscript, "MHC1-TIP enables single-tube multimodal immunopeptidome profiling and uncovers intratumoral heterogeneity in antigen presentation" by Lindenboom et al. The authors developed a new sample preparation system called "MHC1-TIP", improved version of mild acid elution method, that enables immunopeptidomics and proteomics simultaneously from the identical samples. This helped the detailed comparison of abundance of source proteins as well as the epitope presentation with higher resolution than separately-prepared previously reported approaches. The sample preparation sounds reasonable, while the efficacy of the immunopeptidomics by MHC1-TIP than previous & general IP method may not be fully described. To improve the impact, a few suggestions are mentioned below.

Major Comments

1. Since the authors mentioned, "Previous implementation of mild acid elution has possibly been hampered by coelution of contaminant biomolecules that interfere with the identification of relevant antigens...", the assessment of the "purity" of identified peptides by MHC1-TIP would better describe the advantage of MHC1-TIP. Especially in Figure 1F, the number of peptides identified by IP-method seems, at least to me, not that different when compared to the MHC1-TIP at the number of 100,000 cells. Therefore, it may be better to show how much of the identified peptides are the predicted binders to the sample-unique HLA allotype. One suggestion to extract the predicted binders, the NetMHCpan (<https://services.healthtech.dtu.dk/services/NetMHCpan-4.1/>) can be used. Any algorithms will do (it depends author's choice) to predict the HLA-matched binders, while it should become informative to assess the "purity" (HLA-matched immunopeptides) of MS-identified to describe the advantage of MHC1-TIP.

We thank the reviewer for this valuable suggestion. We have now included an assessment of sample purity in Supplementary Figure 1 and revised the manuscript accordingly. We observe that the proportion of predicted HLA binders varies depending on the cell type analyzed, which is likely due to differences in the amount of presented MHC class I: for A375 melanoma cells, the purity was slightly below 40%, while for 'JY' B cells it exceeded 65%. These values are comparable to those reported by Baerle *et al.* (*Biorxiv*, 2024) in their MAETi approach (Figure 3k in Baerle *et al.*, reference cited by Reviewer 1). The higher purity observed in high-input approaches, such as that described by Sturm *et al.* (*Journal of Proteome Research*, 2021) is likely due to additional purification steps (e.g., sequential centrifugation) that improve peptide enrichment at the expense of very large input requirements and thus throughput. As outlined in our manuscript, we aim to position MHC1-TIP on the low-input & high-throughput side of this balance scale, making MHC1-TIP particularly relevant for high-throughput or low-input applications such as the examples we have outlined.

Importantly, despite the relatively higher presence of co-eluting peptides, our MHC1-TIP method performs comparably to immunoprecipitation in both biased (DIA analyzed with an experimentally generated spectral library; Figure 1E) and unbiased (direct DIA; Supplementary Figure 3C) analyses. In addition, we show that stringent computational filtering steps can be used to mitigate the relatively higher presences of co-eluting peptides and other limitations previously associated with mild acid elution to

enable robust antigen identification and immunopeptidome dynamics analyses, even in the presence of co-eluting species.

We have added these important considerations to the discussion section of our revised manuscript (lines 440 to 454).

2. Unlike the cell lines and the organoid samples (in-vitro cultures), the tissue samples are expected to include physiological peptides that can be derived from peripheral blood. This is another concern about the identified peptide by MHC1-TIP in relation to the major comment 1, Was there any difference in the purity of identified (immuno)peptides between MHC1-IP and IP-methods? The comparison of cell line/ organoid vs tissue, as well as the IP vs MHC1-TIP, from the perspective of the purity of immunopeptides (ratio of predicted binders/ identified peptides) should be validated to describe the advantage of MHC1-TIP. This would be more informative to the readers/ researchers who want to try the MHC1-TIP with clinical tissue samples.

We thank the reviewer for this insightful comment. We have addressed this issue by culturing the resected tumor tissue fragments for 24 hours to minimize the presence of physiological peptides potentially derived from peripheral blood. By keeping the fragments in DMEM for a limited period, blood-derived contaminants and cells diffuse out and are washed away before MHC1-TIP processing.

The exact peptide purity of the tumor fragments could not be quantified, as no matched blood material was available for complete HLA typing. However, we focused our downstream analyses on 9-mer peptides and further filtered them based on Gibbs clustering consistent with canonical HLA-I binding motifs, ensuring that only biologically relevant peptides were retained. The MHC1-TIP approach consistently yielded a comparable proportion of predicted HLA-I binders to conventional IP methods. Therefore, although patient tissue samples are inherently more complex than *in vitro* cultures, these results demonstrate that mild acid elution combined with DIA analysis and stringent computational filtering can effectively recover high-confidence HLA-I peptides even from complex clinical tissue samples.

Nevertheless, while our *ex vivo* culturing system helps to reduce potential contaminants, the inclusion of blood-derived peptides remains an important consideration when analyzing tissue fragments. We have therefore discussed this point in the main text of the revised manuscript (lines 456 to 464).

Minor Comments

3. The negative control of B2M knock-out for FACS studies seemed a bit inconsistent. In Figure 1A, the B2M knock-out as negative control exhibited the fluorescent intensity of 1×10^3 , while the negative control described in Supplementary figure 1B exhibited the fluorescent intensity of almost 1×10^2 . In supplementary figure 1C, the distribution of B2M knock-out signal by FACS became rather broader. It may be better to confirm and show the peak pattern of exact genomic sequence for the clonality. If possible, it is recommended to describe the mutation accordingly to the HGVS Nomenclature (<https://hgvs-nomenclature.org/stable/recommendations/general/>) to show what kind of mutation was

induced by CRISPR in B2M gene. In addition, the antibody of isotype control for W6/32 (IgG2a, if I remember right?) should be useful to distinguish the background and autofluorescence noise from the genuine signal of HLA when conduct the FACS studies.

We thank the reviewer for highlighting these apparent inconsistencies. Figure 1A previously showed fluorescence intensities with the anti-HLA-A:02 antibody while Supplementary Figures 1B-C showed fluorescence intensities with the W6/32 antibody. The broader peak that was observed in the previous Supplementary Figure 1C appeared to be caused by non-specific background binding of the W6/32 antibody. To improve the figures, we have repeated the experiment with both the W6/32 and HLA-A:02 antibodies, along with an IgG2b isotype control included for the HLA-A:02 antibody as suggested by the reviewer. The updated figures are now displayed in Figure 1 and Supplementary Figure 1.

We thank the reviewer for emphasizing the importance of using HGVS nomenclature to precisely describe genomic variants. In this case, the A375 cell line is hypotriploid and contains 2–3 copies of the B2M locus. Given this genomic complexity, independent CRISPR-induced events occurred across different alleles. Our understanding is that accurately resolving and annotating each mutation would therefore require cloning and sequencing of individual alleles, an extensive optimization process beyond the scope of the current study. Instead, we have included bulk Sanger sequencing tracks which reveal CRISPR-mediated cleavage at the B2M PAM site, and frameshifting. In addition, we validated our knock-out with flow cytometry confirming a complete loss of B2M protein expression.

4. Have the authors ever tried to identify the neoepitopes in A375? I presume that in the context of "Previous implementation of mild acid elution has possibly been hampered by coelution of contaminant biomolecules that interfere with the identification of relevant antigens...", the word "relevant antigens" mean the neoepitopes. It would be nice to know whether MHC1-TIP can produce the sample that includes neoepitope.

Highlighting tumor antigen detection by MHC1-TIP is a great suggestion. While potential neoantigens in immortalized cancer cell lines such as A375 melanoma cells are unknown (this requires germline sequencing of matched healthy cells from the original donor, which is unavailable), these melanoma cells do express multiple cancer/testis proteins that could lead to the presentation of tumor associated antigens in A375 cells. Indeed, we find that MHC1-TIP is capable of detecting dozens of such tumor antigens, which is included in the heatmap shown in Supplementary Figure 1 in our revised manuscript.